# BDNF produced by cerebral microglia promotes cortical plasticity and pain hypersensitivity after peripheral nerve injury

Lianyan Huang[1,2]*, Jianhua Jin[2], Kai Chen[3], Sikun You[2], Hongyang Zhang[2], Alexandra Sideris[1], Monica Norcini[1], Esperanza Recio-Pinto[1], Jing Wang[1], Wen-Biao Gan[1,4], Guang Yang[1,3]*

1 Department of Anesthesiology, New York University School of Medicine, New York, New York, United States of America, 2 Neuroscience Program, Guangdong Provincial Key Laboratory of Brain Function and Disease, Zhongshan School of Medicine, Sun Yat-sen University, Guangzhou, Guangdong, China, 3 Department of Anesthesiology, Columbia University Medical Center, New York, New York, United States of America, 4 Skirball Institute, Department of Neuroscience and Physiology, New York University School of Medicine, New York, New York, United States of America

* huangly55@mail.sysu.edu.cn (LH); gy2268@cumc.columbia.edu (GY)

## Abstract

Peripheral nerve injury–induced mechanical allodynia is often accompanied by abnormalities in the higher cortical regions, yet the mechanisms underlying such maladaptive cortical plasticity remain unclear. Here, we show that in male mice, structural and functional changes in the primary somatosensory cortex (S1) caused by peripheral nerve injury require neuron-microglial signaling within the local circuit. Following peripheral nerve injury, microglia in the S1 maintain ramified morphology and normal density but up-regulate the mRNA expression of brain-derived neurotrophic factor (BDNF). Using in vivo two-photon imaging and *Cx3cr1*^CreER;*Bdnf*^flox mice, we show that conditional knockout of BDNF from microglia prevents nerve injury–induced synaptic remodeling and pyramidal neuron hyperactivity in the S1, as well as pain hypersensitivity in mice. Importantly, S1-targeted removal of microglial BDNF largely recapitulates the beneficial effects of systemic BDNF depletion on cortical plasticity and allodynia. Together, these findings reveal a pivotal role of cerebral microglial BDNF in somatosensory cortical plasticity and pain hypersensitivity.

## Introduction

Neuropathic pain is caused by lesions and diseases of the somatosensory system and remains one of the most challenging problems in medicine [1]. The primary somatosensory cortex (S1) is critical for sensory processing, and its maladaptive plasticity has been implicated in mediating abnormal sensations associated with neuropathic pain, including the aversion to light touch (mechanical allodynia) [2–4]. Patients and animals under chronic pain states exhibit increased activation and somatotopic reorganization in the S1 [5–7], the extent of which are correlated with pain intensity levels [8,9]. Chronic pain states are also associated with synapse remodeling [10,11], increased pyramidal neuron activity [12,13], and decreased GABAergic

**Funding:** This work was supported by the National Institutes of Health (https://www.nih.gov/) R21NS106469 (G.Y.), R35GM131765 (G.Y.), R01AA027108 (G. Y. and W.-B.G.) and R01GM115384 (J.W.). The funders had no role in study design, data collection and analysis, decision to publish, or preparation of the manuscript.

**Competing interests:** The authors have declared that no competing interests exist.

**Abbreviations:** AUC, area under the curve; BDNF, brain-derived neurotrophic factor; CNS, central nervous system; EYFP, enhanced yellow fluorescent protein; IACUC, Institutional Animal Care and Use Committee; L5, layer 5; MAPK, mitogen-activated protein kinase; NIH, National Institutes of Health; qRT-PCR, quantitative reverse transcriptase PCR; SNI, spared sciatic nerve injury; S1, primary somatosensory cortex; YFP, yellow fluorescent protein; 4-OHT, 4-hydroxytamoxifen.

inhibition in the S1 [13,14]. Furthermore, strategies to reduce cortical changes in the S1 show benefits against chronic pain [11–16]. Despite the importance of S1 in pain processing, the precise mechanisms underlying somatosensory cortical plasticity associated with neuropathic pain remain unclear.

Increasing evidence suggests that immune cells play active roles in neuronal plasticity and chronic pain, and their involvement in pain modulation appears to be sexually dimorphic [17–19]. For example, microglial signaling has been linked to neuropathic pain hypersensitivity in male rodents, while T cells are thought to be involved in females [20]. As resident immune cells, microglia occupy all regions of the mammalian central nervous system (CNS), including brain and spinal cord. Following peripheral nerve injury, microglia in the dorsal horn, the sensory processing region of the spinal cord, transform into reactive phenotypes, producing and releasing a variety of substances that modulate spinal neuron functions. Among those, brain-derived neurotrophic factor (BDNF) has been shown to drive disinhibition and hyperexcitation of dorsal horn neurons, ultimately resulting in pain hypersensitivity [21]. Consistently, pain hypersensitivity in male mice is reduced by depletion of spinal microglia or microglial BDNF [20,22–24].

Despite the crucial role of microglia signaling in spinal mechanisms of chronic pain, the function of brain microglia in pain-related somatosensory plasticity is unclear. In contrast to spinal microglia, microglia in the S1 have been shown to maintain a surveillant morphological phenotype and a normal cell density after peripheral nerve injury [25]. Moreover, the expression of *Bdnf* mRNA is very low in microglia in the adult brain, as indicated by recent RNAseq data [26,27]. Whether cortical microglia increase BDNF expression and contribute to the pathogenesis of chronic pain is not known.

In this study, we investigated the link between cortical microglia, somatosensory cortical plasticity, and pain hypersensitivity in a mouse model of neuropathic pain [28–30]. Using quantitative PCR and RNAscope fluorescence in situ hybridization, we show that microglia in the S1 up-regulate *Bdnf* expression after peripheral nerve injury. This increase of *Bdnf* mRNA is robust and lasting only in males. By performing in vivo two-photon imaging in *Cx3cr1*[CreER/+]:*Bdnf*[fl/fl] male mice, we further show that either systemic or S1-targeted removal of microglial BDNF reduces peripheral nerve injury–induced structural synaptic remodeling and pyramidal neuron hyperactivity in the S1 and alleviates mechanical allodynia in mice. Together, our results demonstrate that BDNF derived from cortical microglia mediates neuronal plasticity in the somatosensory cortex and promotes neuropathic pain hypersensitivity.

## Results

### Microglia in the S1 increase *Bdnf* expression after peripheral nerve injury

To investigate changes of cortical microglia associated with neuropathic pain, we first performed in vivo two-photon imaging in the S1 of 2-month-old male mice expressing enhanced yellow fluorescent protein (EYFP) in microglia under the control of *Cx3cr1* promotor [31]. To induce neuropathic pain, mice were subjected to spared sciatic nerve injury (SNI). Two days after injury, SNI mice exhibited a sustained reduction in the withdrawal threshold of the injured paw upon pressure application to the plantar surface by a von Frey filament, which lasted for >1 month (**Fig 1A**). We examined the morphology, cell density, and process motility of microglia in the S1 contralateral to the SNI side 1 week after surgery. Consistent with previous reports [25], we observed no differences in the morphology and density of microglia in the S1 between SNI and sham mice (**Fig 1B and 1C**). However, we found that the processes of microglia were less motile in SNI mice as compared to sham mice (**Fig 1D and 1E**).

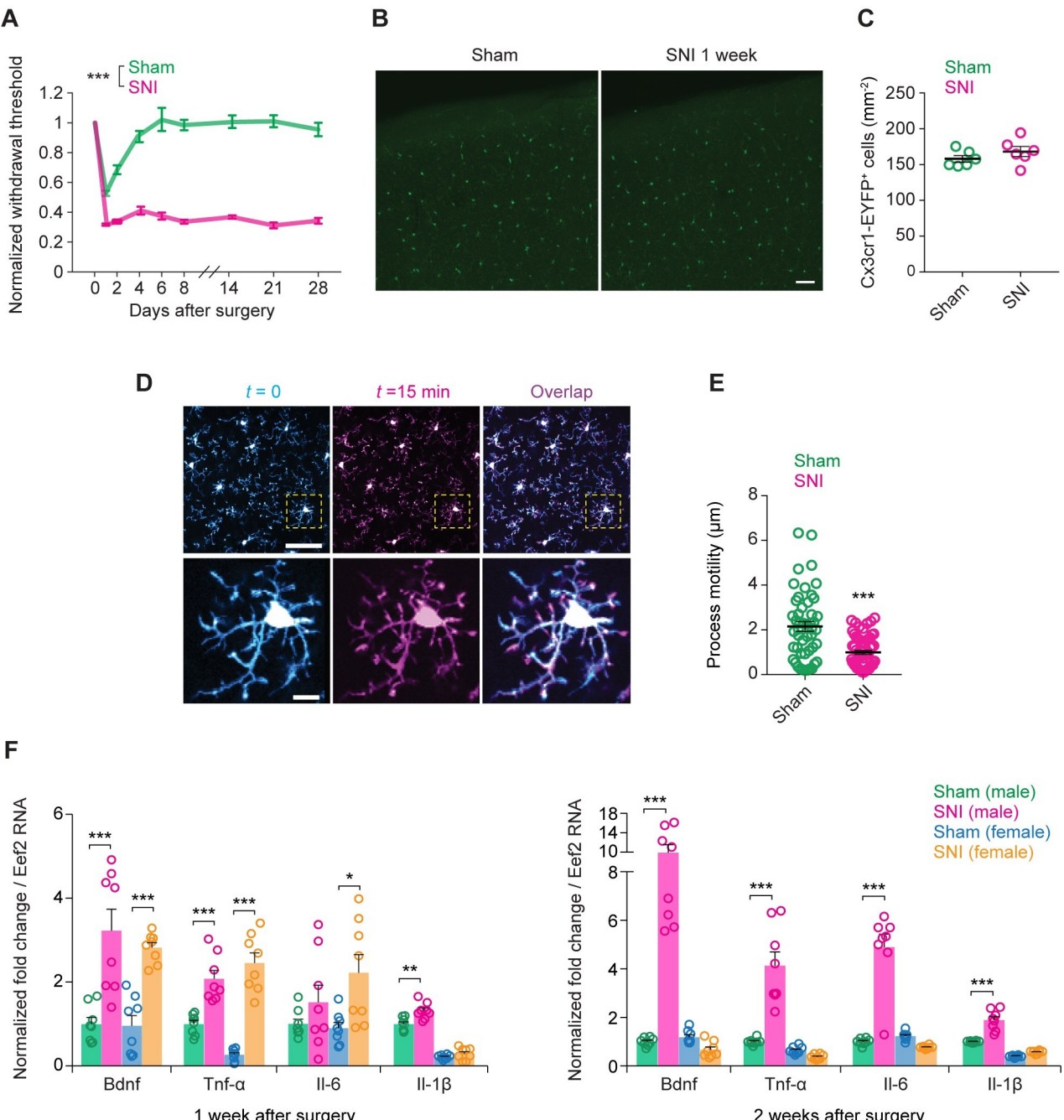

**Fig 1. Microglia maintain normal morphology and density in the S1 after peripheral nerve injury but increase gliotransmitter expression.** (**A**) Paw withdrawal threshold before and after SNI ($n = 14$ mice) or sham ($n = 10$ mice) surgery ($F_{1,22} = 569.6$, $P < 0.001$). (**B**) Coronal sections of S1 from 2- to 3-month-old *Cx3cr1*$^{CreER-EYFP}$ mice stained for EYFP. Scale bar, 50 μm. (**C**) Quantification of microglia (*Cx3cr1*-EYFP$^+$ cell) density in the S1 of sham and SNI mice 1 week after surgery ($t_{10} = 1.206$, $P = 0.2556$; $n = 6$ mice per group). Individual circle represents data from a single animal. (**D**) Motility of microglial process in the S1 was measured by intravital two-photon imaging 1 week after surgery. Scale bar, 50 μm (top) and 10 μm (bottom). (**E**) Quantification of S1 microglial process extensions and retractions over 15 min in sham and SNI mice ($t_{107} = 5.196$, $P < 0.0001$; $n = 6$ mice per group). Individual circle represents data from a single cell. (**F**) *Bdnf*, *Tnf-α*, *Il-6*, and *Il-1β* mRNA expression in S1 microglia 1 week and 2 weeks after surgery ($n = 8$ mice per group). Summary data are presented as mean ± SEM. $^*P < 0.05$, $^{**}P < 0.01$, $^{***}P < 0.001$; by an unpaired *t* test (C, E), two-way ANOVA followed by Bonferroni multiple comparisons test (A), or one-way ANOVA followed by Bonferroni multiple comparisons test (F). (B, D) Representative images from experiments carried out on at least 5 animals per group. The data underlying this figure can be found in S1 Data. EYFP, enhanced yellow fluorescent protein; SNI, spared sciatic nerve injury; S1, primary somatosensory cortex.

Next, we isolated microglia from the S1 using fluorescence-activated cell sorting and measured the mRNA levels of *Bdnf* and various cytokines. Quantitative PCR results indicate that SNI in male mice markedly increased *Bdnf* mRNA expression by 3- and 10-fold over 1 and 2 weeks, respectively (**Fig 1F**). The mRNA levels of *Tnf-α*, *Il-6*, and *Il-1β* in microglia were also significantly elevated in males 2 weeks after SNI. By contrast, female mice exhibited a transient elevation of *Bdnf*, *Tnf-α*, and *Il-6* transcripts in microglia (**Fig 1F**). Two weeks after surgery, there was no difference in the mRNA expression of various gliotransmitters between SNI and sham groups in females. These results indicate that cortical microglia respond to peripheral nerve injury in a sex-dependent manner, consistent with the notion that the involvement of microglial signaling in the development of chronic pain is sexually dimorphic.

To confirm peripheral nerve injury–induced increases of *Bdnf* mRNA expression in cortical microglia, we used RNAscope fluorescence in situ hybridization to visualize *Bdnf* transcripts in brain sections. To do this, we utilized predesigned *Bdnf* RNAScope probes targeting *Bdnf* open reading frame (**Fig 2A**). To ensure their specificity, positive control probes targeting the housekeeping gene *Polr2a* and negative control probes targeting *DapB* were run alongside the *Bdnf* probes (**Fig 2B and 2C**). Once the specificity of the probes was confirmed, both contralateral and ipsilateral S1 were imaged from brain sections that were taken from mice in sham (*n* = 4) and SNI (*n* = 4) groups. As shown in **Fig 2A**, *Bdnf* mRNA was readily detectable in cortical slices. Consistent with the notion that neurons are the major source of BDNF, only a small fraction of *Bdnf* transcripts were colocalized with EYFP-labeled microglia. In line with previous studies [26], we found that *Bdnf* mRNA expression was very low in microglia under physiological conditions. In sham mice, *Bdnf* mRNA was detectable in only approximately 30% microglial cells (**Fig 2A and 2D**, **S1 Fig**). Among those positive cells, on average, 2 *Bdnf* mRNA puncta were identified per microglia (**Fig 2A and 2E**, **S1 Fig**). As compared to sham, SNI significantly increased the fraction of microglial cells expressing *Bdnf* mRNA, as well as the number of *Bdnf* mRNA puncta per microglia, in contralateral, but not ipsilateral S1 (**Fig 2D and 2E**). Together, these results indicate that cortical microglia up-regulate *Bdnf* mRNA expression after peripheral nerve injury.

## Genetic depletion of microglial BDNF prevents injury-induced synapse remodeling

Previous studies in mice have shown that peripheral nerve injury elicits a rapid structural synaptic remodeling in the S1 within days [10]. To determine whether microglia-derived BDNF is critical for synaptic plasticity associated with chronic neuropathic pain in male mice, we examined the dynamics of postsynaptic dendritic spines in the S1 with transcranial two-photon microscopy. Two-month-old *Thy1*-YFP transgenic mice expressing yellow fluorescent protein (YFP) in layer 5 (L5) pyramidal neurons were repeatedly imaged before and 1 to 4 weeks after peripheral nerve injury (**Fig 3A**). In the region of S1 corresponding to the SNI surgery side, we found a significant increase in the elimination and formation rates of dendritic spines (**Fig 3B and 3C**). Within the first week after surgery, 7.7 ± 0.6% of dendritic spines were eliminated and 8.1 ± 0.5% were formed in sham mice, whereas 11.4 ± 1.1% and 11.9 ± 1.1% of dendritic spines were eliminated and formed in SNI mice, respectively. This abnormally high turnover of dendritic spines occurred in the S1 contralateral to the surgery site, but not in the ipsilateral S1 or in the visual cortex, a cortical area unrelated to pain perception (**Fig 3C**). There was a significant correlation between the percentages of dendritic spines eliminated and formed on individual dendrites in SNI mice (**Fig 3D**), and the total spine number in the S1 remained the same (**S2 Fig**). Furthermore, 2 to 4 weeks after SNI, the rates of spine elimination and

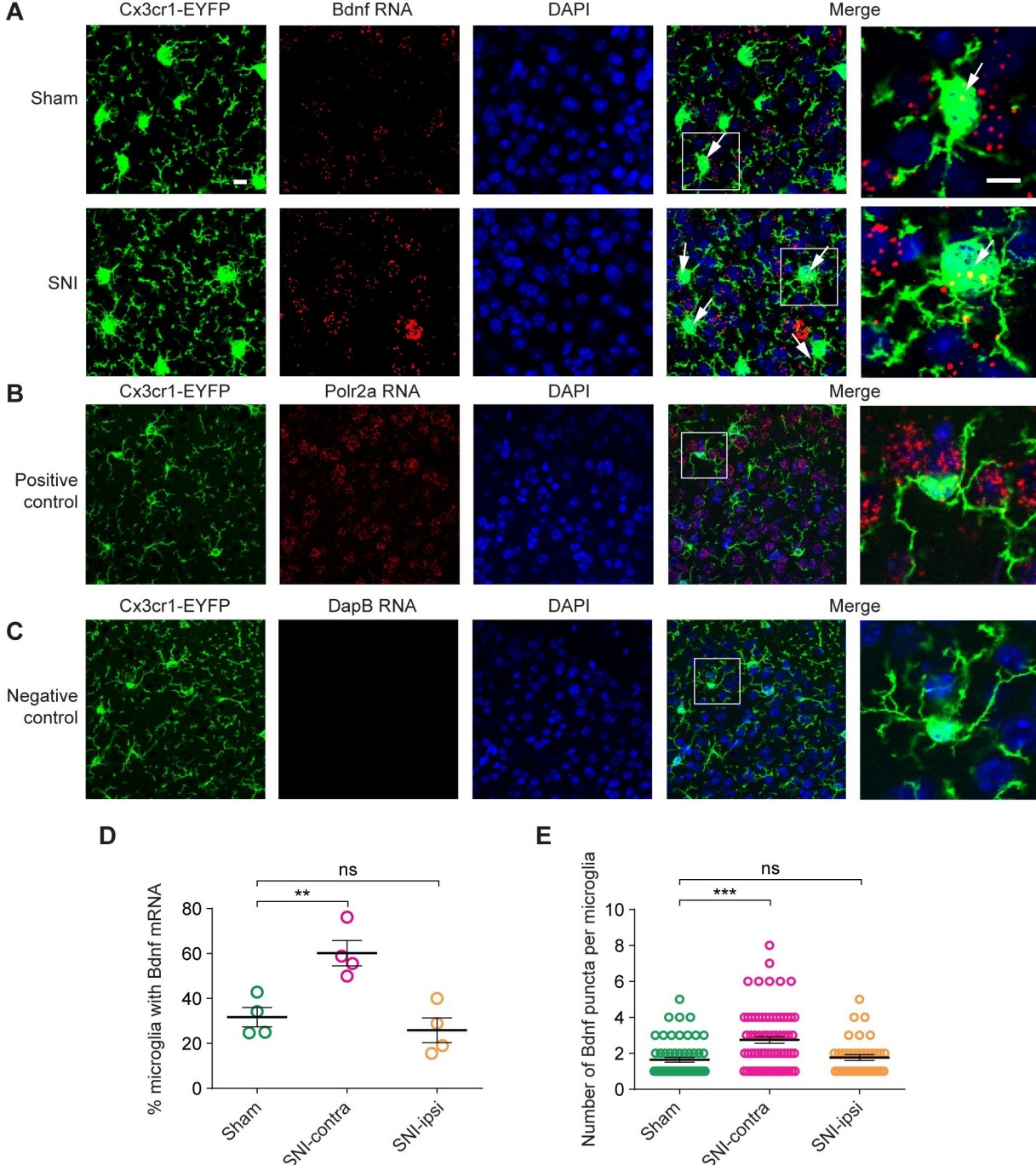

**Fig 2. Validation of microglial *Bdnf* through RNAscope fluorescence in situ hybridization.** (**A**) RNAscope fluorescence in situ hybridization in the S1. Red color represents *Bdnf* mRNA probe hybridization. Green color indicates *Cx3cr1*-EYFP⁺ microglia. Blue, DAPI. Arrows indicate *Bdnf* transcripts located within microglia. Scale bar, 10 μm. (**B**) Positive control probes for in situ hybridization, targeting mRNA for the ubiquitously expressed housekeeping gene *Polr2a*; images show representative signal in S1. (**C**) Negative control probes targeting mRNA for *DapB*, a gene expressed in *Bacillus subtilis*. (**D**) Percentages of microglia with the presence of *Bdnf* mRNA ($n$ = 4 mice per group, $F_{2, 9}$ = 12.49, $P$ = 0.0025). (**E**) Number of *Bdnf* puncta per cell among microglia with detectable *Bdnf* transcripts ($F_{2, 168}$ = 13.63, $P < 0.0001$). Summary data are presented as mean ± SEM. **$P < 0.01$, ***$P < 0.001$, ns, not significant; by one-way ANOVA followed by Tukey multiple comparisons test (**D**, **E**). (**A-C**) Representative images from experiments carried out on 4 animals per group. The data underlying this figure can be found in S1 Data. *Bdnf*, brain-derived neurotrophic factor; SNI, spared sciatic nerve injury; S1, primary somatosensory cortex.

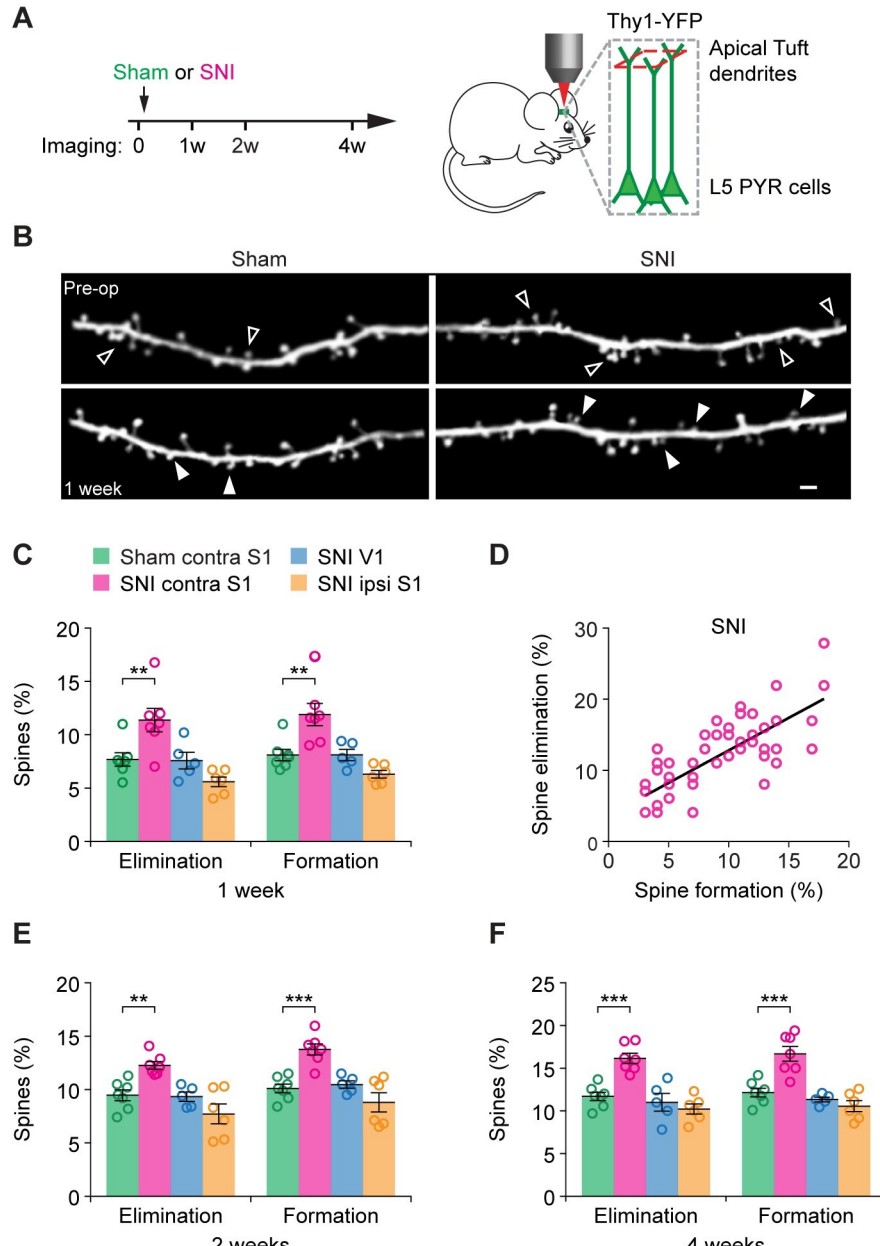

**Fig 3. Persistent dendritic spine remodeling in the S1 after peripheral nerve injury.** (**A**) Schematic showing the timeline for transcranial two-photon imaging and SNI or sham operation. Imaging was performed in transgenic mice expressing YFP in L5 PYR neurons. (**B**) Representative two-photon images of dendritic spines on apical dendritic segments of L5 pyramidal neurons in the S1 of SNI and sham mice. Empty and filled arrow heads indicate spines that were eliminated or newly formed, respectively, on the same dendritic segment. Scale bar, 2 μm. (**C**) Percentages of dendritic spines eliminated and formed in various cortical areas 1 week after SNI or sham surgery (Elimination: $F_{3, 21}$ = 9.455, $P$ = 0.0004; Formation: $F_{3, 21}$ = 11.54, $P$ = 0.0001). Sham contra S1, 1,074 spines, $n$ = 7 mice; SNI contra S1, 1,106 spines, $n$ = 7 mice; SNI V1, 716 spines, $n$ = 5 mice, SNI ipsi S1, 874 spines, $n$ = 6 mice. (**D**) Correlation between the rates of spine elimination and spine formation 1 week after SNI (Pearson $r$ = 0.77, $P$ < 0.0001). (**E**) Percentages of dendritic spine elimination and formation 2 weeks after surgery (Elimination: $F_{3, 21}$ = 10.7, $P$ = 0.0002; Formation: $F_{3, 21}$ = 13.84, $P$ < 0.0001). (**F**) Percentages of dendritic spine elimination and formation 4 weeks after surgery (Elimination: $F_{3, 21}$ = 17.25, $P$ < 0.0001; Formation: $F_{3, 21}$ = 18.49, $P$ < 0.0001). Individual circles represent data from a single mouse. Summary data are presented as mean ± SEM. **$P$ < 0.01, ***$P$ < 0.001; by one-way ANOVA followed by Tukey multiple comparisons test (**C, E, F**). (**B**) Representative images from experiments carried out on at least 5 animals per group. The data underlying this figure can be found in S1 Data. L5, layer 5; PYR, pyramidal; SNI, spared sciatic nerve injury; S1, primary somatosensory cortex; YFP, yellow fluorescent protein.

formation remained elevated in comparison to sham mice (**Fig 3E and 3F**), indicating persistent synaptic remodeling in the S1 of mice with chronic neuropathic pain.

To determine whether injury-induced dendritic spine remodeling involves microglial BDNF, we performed in vivo gene deletion by crossing *Cx3cr1*[CreER] mice with mice containing 1 or 2 floxed alleles of *Bdnf* (*Bdnf*[fl/+] *or Bdnf*[fl/fl]) [32,33] (**Fig 4**). Specifically, *Cx3cr1*[CreER/+]; *Bdnf*[fl/+];*Thy1*-YFP or *Cx3cr1*[CreER/+];*Bdnf*[fl/fl];*Thy1*-YFP males were given 2 doses of tamoxifen (P30 and 32). Tamoxifen-induced gene recombination removes one or both copies of the *Bdnf* gene from CX3CR1-expressing microglia [32]. As expected, no *Bdnf* mRNA was detected in the soma of microglia in tamoxifen-treated *Cx3cr1*[CreER/+];*Bdnf*[fl/fl] mice (**S3 Fig**). Interestingly, SNI-induced up-regulation of *Tnf*-α expression in microglia was also reduced in these mice depleted of microglial BDNF (**S4 Fig**). One month after the last tamoxifen treatment (i.e., P60), we performed SNI or sham surgery and examined the rates of dendritic spine elimination and formation in the S1 over the next 1 to 4 weeks (**Fig 4A**). Similar to wild-type mice, we found a marked increase in spine elimination and formation 1 week after SNI, in *Cx3cr1*[CreER/+];*Bdnf*[fl/+] (hereafter referred to as *Bdnf*[fl/+]) mice (**Fig 4B and 4C**). In contrast, in *Cx3cr1*[CreER/+]; *Bdnf*[fl/fl] mice that removed both copies of the *Bdnf* gene from microglia (hereafter referred to as *Bdnf*[fl/fl]), we found that SNI had no significant effects on the rates of dendritic spine elimination as compared to sham (6.8 ± 0.5% versus 5.4 ± 0.4%, P = 0.0747; **Fig 4B and 4C**). In *Bdnf*[fl/fl] mice, there was a slight but significant increase of spine formation after SNI in comparison to sham (**Fig 4C**). However, the degree of SNI-induced spine formation in *Bdnf*[fl/fl] mice was

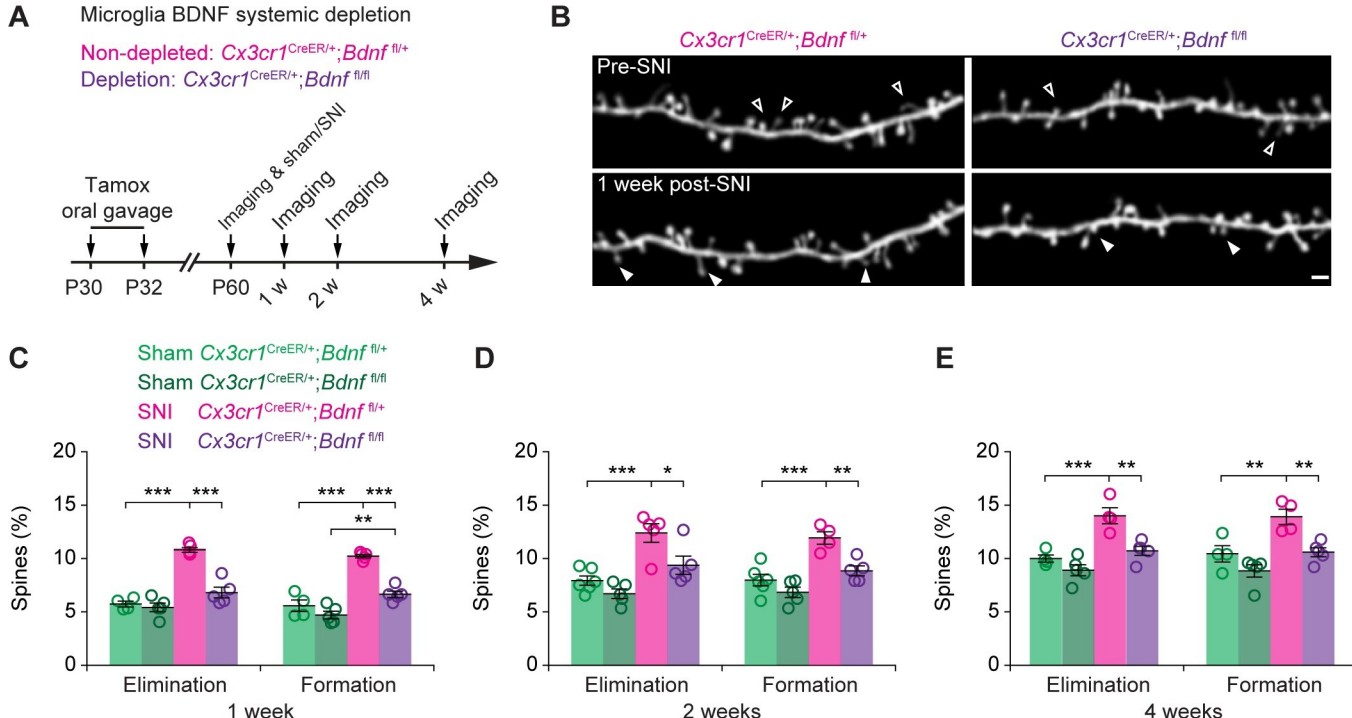

**Fig 4. Systemic depletion of microglial BDNF reduces dendritic spine remodeling in the S1 after peripheral nerve injury.** (**A**) Experimental timeline. (**B**) Representative two-photon images of dendritic segments in SNI mice with (*Cx3cr1*[CreER/+];*Bdnf*[fl/+]) or without microglial BDNF (*Cx3cr1*[CreER/+];*Bdnf*[fl/fl]). Empty and filled arrow heads indicate individual spines that were eliminated or newly formed, respectively, 1 week after SNI. Scale bar, 2 μm. (**C–E**) Percentages of dendritic spines eliminated and formed 1 week (**C**), 2 weeks (**D**), or 4 weeks (**E**) after sham or SNI surgery. Sham *Bdnf*[fl/+]: n = 4–6 mice; SNI *Bdnf*[fl/+], n = 4–5 mice; sham *Bdnf*[fl/fl], n = 5 mice; SNI *Bdnf*[fl/fl], n = 5 mice. Throughout, individual circles represent data from a single mouse. Summary data are presented as mean ± SEM. *P < 0.05, **P < 0.01, ***P < 0.001; by one-way ANOVA followed by Tukey multiple comparisons test (**C–E**). The data underlying this figure can be found in S1 Data. BDNF, brain-derived neurotrophic factor; SNI, spared sciatic nerve injury; S1, primary somatosensory cortex.

significantly lower than that in $Bdnf^{fl/+}$ mice ($P < 0.0001$). We also found that after SNI surgery, the rates of spine elimination and formation over 2 to 4 weeks were lower in $Bdnf^{fl/fl}$ mice than that in $Bdnf^{fl/+}$ mice (**Fig 4D and 4E**). Together, these results indicate that depletion of BDNF from microglia reduces dendritic spine remodeling in the S1 after peripheral nerve injury.

## Genetic depletion of microglial BDNF prevents neuronal hyperactivation and mechanical allodynia

Next, we investigated the effect of microglial BDNF depletion on cortical activity after peripheral nerve injury. We first performed in vivo two-photon $Ca^{2+}$ imaging in the somas of L5 pyramidal neurons expressing a genetically encoded $Ca^{2+}$ indicator GCaMP6s in the S1 of awake, head-restrained wild-type mice (**Fig 5A**) [34]. Consistent with previous studies [13], we found that the spontaneous $Ca^{2+}$ activity of S1 neurons was significantly higher in SNI mice than that in sham mice 1 week after surgery (**Fig 5B–5D**), indicating somatosensory cortical activation in mice with neuropathic pain. Mechanical stimulation of the hind paw using a 0.6-g von Frey hair elicited a marked increase of somatic $Ca^{2+}$ responses in L5 neurons in SNI mice, but not in sham mice (**Fig 5E and 5F**). Furthermore, more L5 cells showed increased $Ca^{2+}$ responses in SNI mice as compared with the sham group (**Fig 5G**). When stronger stimulation was delivered through a 1- or 2-g von Frey hair, increases in neuronal activity were observed in both SNI and sham mice in comparison to the resting condition (**S5 Fig**). With the same stimuli, the level of stimulation evoked $Ca^{2+}$ in L5 somas was significantly higher in SNI mice than in sham mice (**Fig 5H**). In addition, 2 weeks after SNI surgery, both spontaneous and evoked $Ca^{2+}$ activity in S1 L5 neurons remained elevated as compared to sham mice (**Fig 5I**, **S5 Fig**). Together, these data indicate that mice with increased mechanical pain sensitivity exhibit a persistent elevation of pyramidal neuronal activity in the S1.

To determine whether microglial BDNF depletion affects injury-induced pyramidal neuronal hyperactivity, we crossed $Thy1$-GCaMP6s mice with $Cx3cr1^{CreER/+}$;$Bdnf^{fl/fl}$ mice and administered 2 doses of tamoxifen on P30 and P32 by oral gavage (**Fig 6A**). One month after the last tamoxifen treatment, we performed SNI surgery and examined L5 pyramidal neuron activity in the S1. In contrast to the hyperactivation of pyramidal neurons in $Cx3cr1^{CreER/+}$; $Bdnf^{fl/+}$ mice, we found that both spontaneous and sensory-evoked neuronal activity in mice depleted of microglial BDNF ($Cx3cr1^{CreER/+}$;$Bdnf^{fl/fl}$) were reduced 1 week after SNI (**Fig 6B and 6C**). Two weeks after SNI, $Bdnf^{fl/fl}$ mice continued to show lower activity in L5 cells as compared to $Bdnf^{fl/+}$ mice (**Fig 6D**). These data indicate that depletion of microglial BDNF is effective in mitigating peripheral nerve injury–induced S1 activation.

When paw withdrawal threshold was measured over time, we found that removal of BDNF from microglia attenuated mechanical allodynia in SNI mice as compared to nondeleted controls (**Fig 6E**). Four weeks after SNI, the withdrawal threshold of $Bdnf^{fl/fl}$ mice remained higher than that of $Bdnf^{fl/+}$ controls, indicating that microglial BDNF deficiency attenuates pain hypersensitivity over long term.

## S1-targeted gene recombination in microglia

The results above demonstrate that systemic depletion of BDNF from CNS microglia (both brain and spinal cord) reduces synaptic plasticity and neuronal hyperactivity in the S1 after peripheral nerve injury. Because spinal microglial BDNF is important for spinal neuronal plasticity and neuropathic pain in male mice [20,21], the effects of systemic BNDF depletion on somatosensory cortical plasticity could be due to the removal of microglial BDNF either in the spinal cord and/or in the cortex. To directly test the role of cortical microglial BDNF in nerve

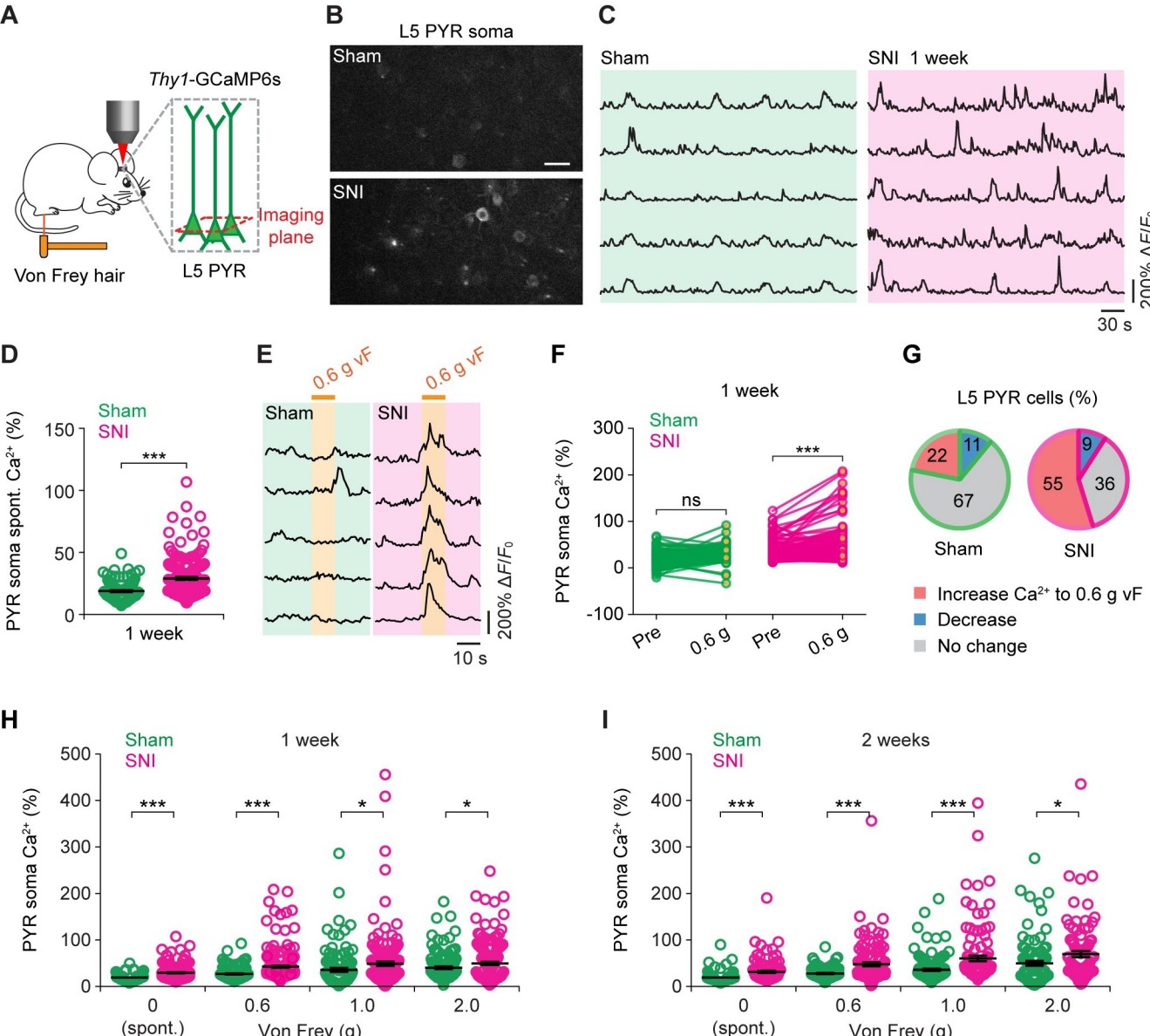

**Fig 5. Enhanced sensory-evoked pyramidal neuronal activity in the S1 after peripheral nerve injury.** (**A**) Cartoon depicting two-photon $Ca^{2+}$ imaging in awake, head-restrained mice expressing GCaMP6s in L5 PYR neurons. (**B**) Representative two-photon images of L5 somas in the S1 of sham and SNI mice 1 week after surgery. Scale bar, 20 μm. (**C**) Representative fluorescence traces of L5 somas in SNI and sham mice 1 week after surgery. (**D**) Spontaneous $Ca^{2+}$ activity of L5 somas over 2.5 min 1 week after surgery (sham: $18.26 \pm 0.73\%$, $n = 115$ cells from 5 mice; SNI: $28.57 \pm 1.18\%$, $n = 179$ cells from 7 mice; $t_{292} = 6.501$, $P < 0.001$). (**E**) Representative fluorescence traces of L5 somas in SNI and sham mice in response to mechanical stimulation applied by a 0.6-g vF hair. (**F**) Averaged $Ca^{2+}$ activity over 10 s before or during 0.6-g vF stimulation (sham: $20.80 \pm 1.52\%$, $24.57 \pm 1.47\%$, $P = 0.123$; SNI: $27.55 \pm 1.27\%$, $41.67 \pm 2.17\%$, $P < 0.001$). (**G**) Percentages of L5 somas showing increased $Ca^{2+}$ responses to 0.6-g stimuli in sham ($n = 115$ cells) and SNI ($n = 179$ cells) mice ($P < 0.01$, chi-squared test). $Ca^{2+}$ activity was analyzed over 10 s. (**H**) vF stimulation–evoked somatic $Ca^{2+}$ in L5 neurons 1 week after surgery (sham: $n = 5$ mice; SNI: $n = 7$ mice; 0 g: $t_{292} = 6.501$, $P < 0.001$; 0.6 g: $t_{329} = 4.963$, $P < 0.001$; 1.0 g: $t_{265} = 1.981$, $P < 0.05$; 2.0 g: $t_{276} = 2.028$, $P < 0.05$). (**I**) vF stimulation–evoked somatic $Ca^{2+}$ in L5 neurons 2 weeks after surgery (sham: $n = 5$ mice; SNI: $n = 5$ mice; 0 g: $t_{211} = 4.920$, $P < 0.001$; 0.6 g: $t_{238} = 4.983$, $P < 0.001$; 1.0 g: $t_{245} = 4.124$, $P < 0.001$; 2.0 g: $t_{190} = 2.421$, $P < 0.05$). Throughout, individual circles represent data from a single cell. Data are presented as mean ± SEM. $^*P < 0.05$, $^{**}P < 0.01$, $^{***}P < 0.001$; by unpaired $t$ test (**D, H, I**), paired $t$ test (**F**). (**B, C, E**) Representative images and traces from experiments carried out on at least 5 animals per group. The data underlying this figure can be found in S1 Data. L5, layer 5; PYR, pyramidal; SNI, spared sciatic nerve injury; S1, primary somatosensory cortex; vF, von Frey.

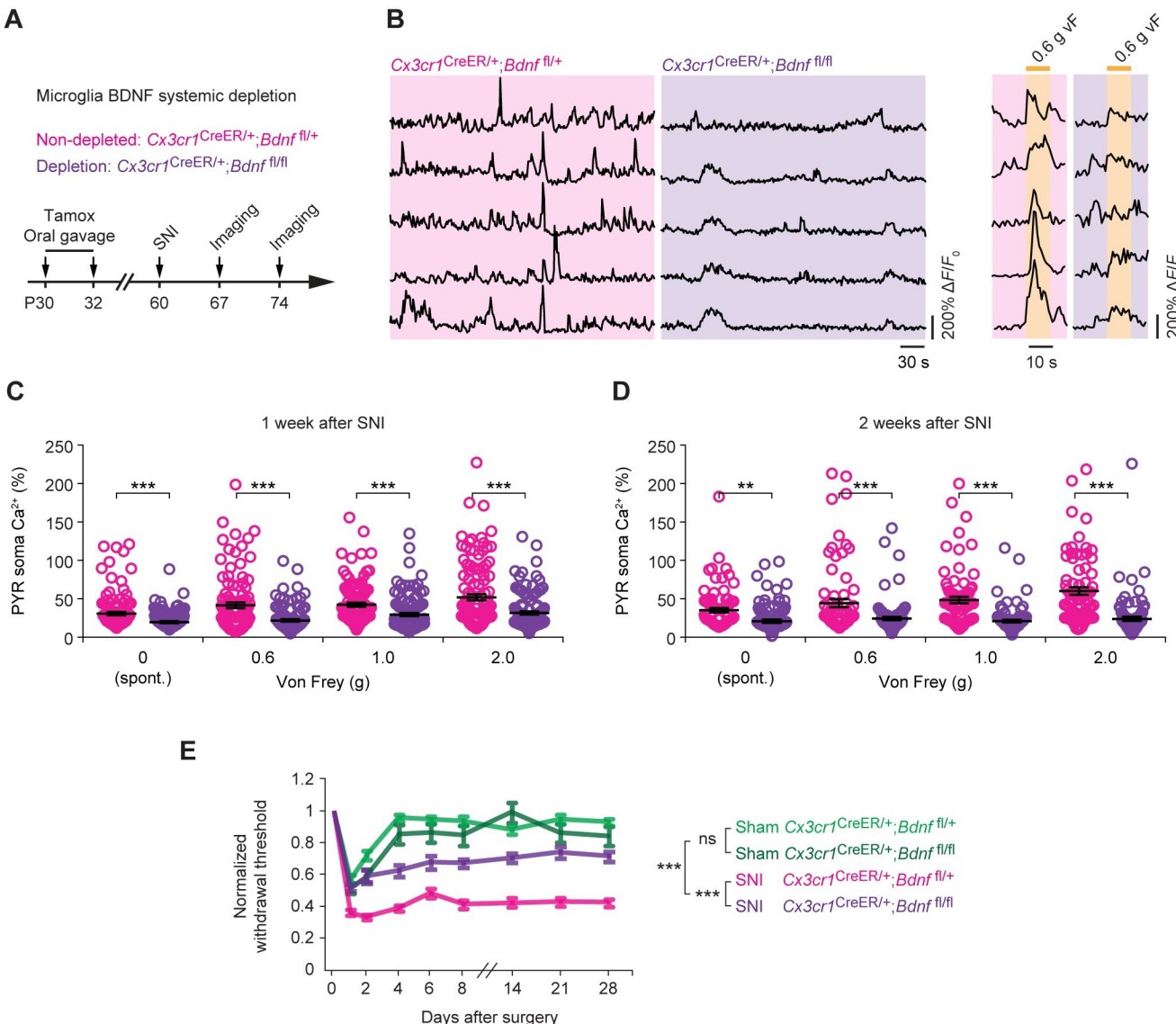

**Fig 6. Systemic depletion of microglial BDNF reduces pyramidal neuronal hyperactivity and mechanical allodynia.** (**A**) Experimental timeline. (**B**) Representative fluorescence traces of L5 PYR somas expressing GCaMP6s in SNI mice with ($Cx3cr1^{CreER/+};Bdnf^{fl/+}$) or without ($Cx3cr1^{CreER/+};Bdnf^{fl/fl}$) microglial BDNF. Images were collected 1 week after surgery when mice were at rest or subjected to 0.6-g vF stimulation. (**C**) Averaged $Ca^{2+}$ activity in L5 somas 1 week after SNI ($Bdnf^{fl/+}$, $n = 4$ mice; $Bdnf^{fl/fl}$, $n = 5$ mice; 0 g: $t_{246} = 3.551$, $P < 0.001$; 0.6 g: $t_{220} = 5.143$, $P < 0.001$; 1.0 g: $t_{232} = 3.986$, $P < 0.001$; 2.0 g: $t_{244} = 4.751$, $P < 0.001$). (**D**) Averaged $Ca^{2+}$ activity in L5 somas 2 weeks after SNI ($Bdnf^{fl/+}$, $n = 5$ mice; $Bdnf^{fl/fl}$, $n = 4$ mice; 0 g: $t_{171} = 2.666$, $P = 0.0084$; 0.6 g: $t_{177} = 3.913$, $P < 0.001$; 1.0 g: $t_{179} = 6.709$, $P < 0.001$; 2.0 g: $t_{174} = 6.918$, $P < 0.001$). (**E**) Paw withdraw threshold in sham and SNI mice with or without microglial BDNF (sham $Bdnf^{fl/+}$, $n = 7$ mice; SNI $Bdnf^{fl/+}$, $n = 9$ mice; sham $Bdnf^{fl/fl}$, $n = 9$ mice, SNI $Bdnf^{fl/fl}$, $n = 8$ mice). (**B**) Representative traces from experiments carried out on at least 4 animals per group. Throughout, individual circle represents data from a single cell. Summary data are presented as means ± SEM. **$P < 0.01$, ***$P < 0.001$; by unpaired $t$ test (**C**, **D**) or two-way ANOVA followed by Bonferroni multiple comparisons test (**E**). The data underlying this figure can be found in S1 Data. BDNF, brain-derived neurotrophic factor; L5, layer 5; ns, not significant; PYR, pyramidal; SNI, spared sciatic nerve injury; vF, von Frey.

injury–induced neuronal plasticity, we developed a strategy to selectively manipulate microglia within the S1 (**Fig 7**). In this experiment, we injected 4-hydroxytamoxifen (4-OHT), the active metabolite of tamoxifen, into the S1 to induce CreER-mediated recombination in a spatially confined manner (**Fig 7A**). To validate the region specificity of gene recombination, we crossed $Cx3cr1^{CreER-EYFP}$ mice to $Rosa26$-stop-DsRed reporter mice ($R26^{DsRed}$). When 4-OHT

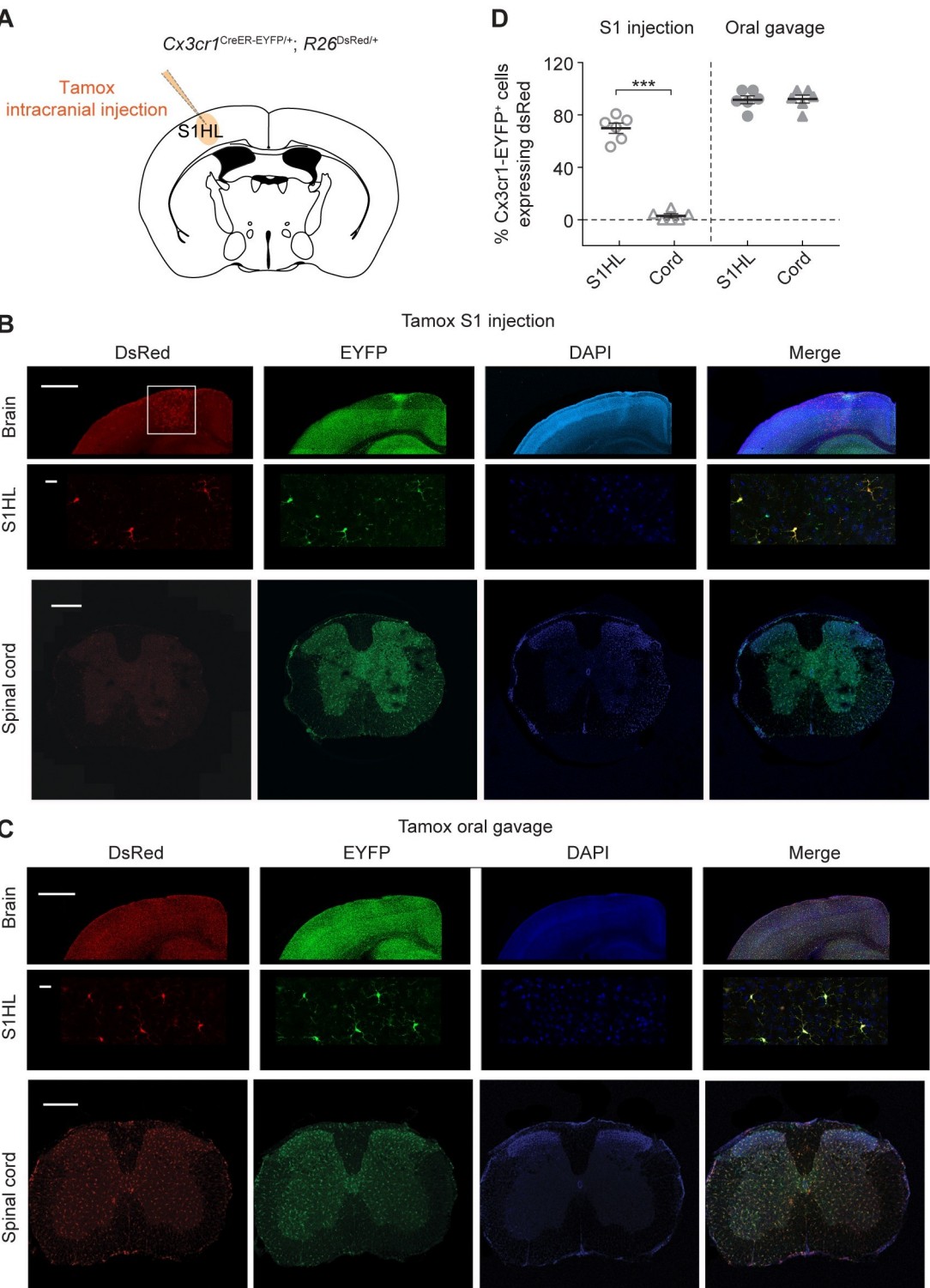

**Fig 7. Strategy to selectively manipulate microglia in S1.** (**A**) Cartoon depicting the strategy to selectively manipulate microglia in the S1HL. (**B**) Representative coronal sections of brain and spinal cord in $Cx3cr1^{CreER-EYFP/+}$;$R26^{DsRed/+}$ mice 1 month after intracranial injection of 4-OHT into the S1HL. Scale bar, top, 1,000 μm; middle, 20 μm; bottom, 1,000 μm. (**C**) Representative coronal sections of brain and spinal cord of $Cx3cr1^{CreER-EYFP/+}$; $R26^{DsRed/+}$ mice after systemic administration of tamoxifen by oral gavage. (**D**) Quantification of $Cx3cr1$-EYFP$^+$ cells expressing DsRed in S1HL and spinal cord after systemic (oral gavage) or local (intracranial injection) administration of tamoxifen ($n$ = 6 mice per group). Summary data are presented as means ± SEM. ***$P$ < 0.001 by unpaired $t$ test. The data underlying this figure can be found in S1 Data. BDNF, brain-derived neurotrophic

factor; EYFP, enhanced yellow fluorescent protein; S1, primary somatosensory cortex; S1HL, hindlimb region of S1; 4-OHT, 4-hydroxytamoxifen.

was locally delivered to the S1 hindlimb region, a large fraction (70.7% ± 3.9%) of *Cx3cr1*-EYFP$^+$ microglia in this region (within 1 mm of the injection site) were found to coexpress DsRed. Importantly, essentially none of the microglia in the spinal cord were DsRed$^+$ (**Fig 7B and 7D**). On the other hand, in *Cx3cr1*$^{CreER/+}$;*R26*$^{DsRed/+}$ mice subjected to systemic administration (oral gavage) of tamoxifen, we found that the majority (92.7% ± 3.0%) of cortical and spinal microglia were DsRed$^+$ 4 weeks later (**Fig 7C and 7D**). Thus, intracranial injection of tamoxifen efficiently and region specifically induced CreER-mediated gene recombination in the cortex of *Cx3cr1*$^{CreER/+}$ mice.

## Cortical microglial BDNF mediates S1 plasticity and mechanical allodynia

Next, we intracranially administered 4-OHT into the contralateral S1 of *Cx3cr1*$^{CreER/+}$;*Bdnf*$^{fl/fl}$ and *Cx3cr1*$^{CreER/+}$;*Bdnf*$^{fl/+}$ male mice. Four weeks later, we performed SNI surgery and examined dendritic spine remodeling, neuronal Ca$^{2+}$ activity, and the animals' mechanical paw withdrawal threshold over time (**Fig 8A**). Similar to what was observed in mice systemically depleted of microglial BDNF, we found that depletion of microglial BDNF within S1 significantly reduced the rates of dendritic spine formation and elimination 1 to 2 weeks after SNI (**Fig 8B and 8C**). Furthermore, SNI-induced neuronal hyperactivity, both spontaneous and sensory evoked, was substantially reduced in mice with cortical BDNF depletion as compared to nondepleted controls (**Fig 8D–8F**). Mechanical allodynia was also reduced in mice locally depleted of BDNF relative to nondepleted controls (**Fig 8G**). Thus, loss of microglial BDNF within the contralateral S1 largely recapitulates the effects on cortical plasticity and pain behavior observed in mice depleted of microglial BDNF in the entire CNS. These results indicate that cortical microglia-derived BDNF is crucial for cortical plasticity associated with neuropathic pain.

## Discussion

Abnormal cortical plasticity is thought to be critical for chronic pain development after peripheral nerve injury, but the underlying mechanisms remain unclear. Here, we show that cortical microglia mediate neuropathic pain in male mice by promoting BDNF-dependent somatosensory cortical plasticity. Depletion of BDNF from microglia either across the entire CNS or specifically within the S1 reduces dendritic spine remodeling and neuronal hyperactivity in cortical pyramidal neurons, as well as mechanical allodynia in male mice. These findings underscore the important role of BDNF derived from cortical microglia in the development of neuropathic pain in males.

Recent findings have revealed that the contributing role of microglia in chronic pain is male biased [35,36]. In the spinal cord, activating Toll-like receptor 4, which is primarily expressed by microglia, by the agonist lipopolysaccharide, causes pain behavior in male but not female mice [37]. In rodent models of chronic inflammatory and neuropathic pain, there is a male-specific activation of microglial signaling in the spinal cord, which includes the up-regulation of P2X4 receptors [38], phosphorylation of p38−mitogen-activated protein kinase (MAPK) [39], and subsequent synthesis and release of BDNF [20]. Whether such male-biased microglial pathways also exist in the brain, contributing to pain hypersensitivity, is less investigated. In the present study, we showed at the mRNA level that microglia in the S1 displayed a lasting increase of gliotransmitters after peripheral nerve injury. Two weeks after injury, the

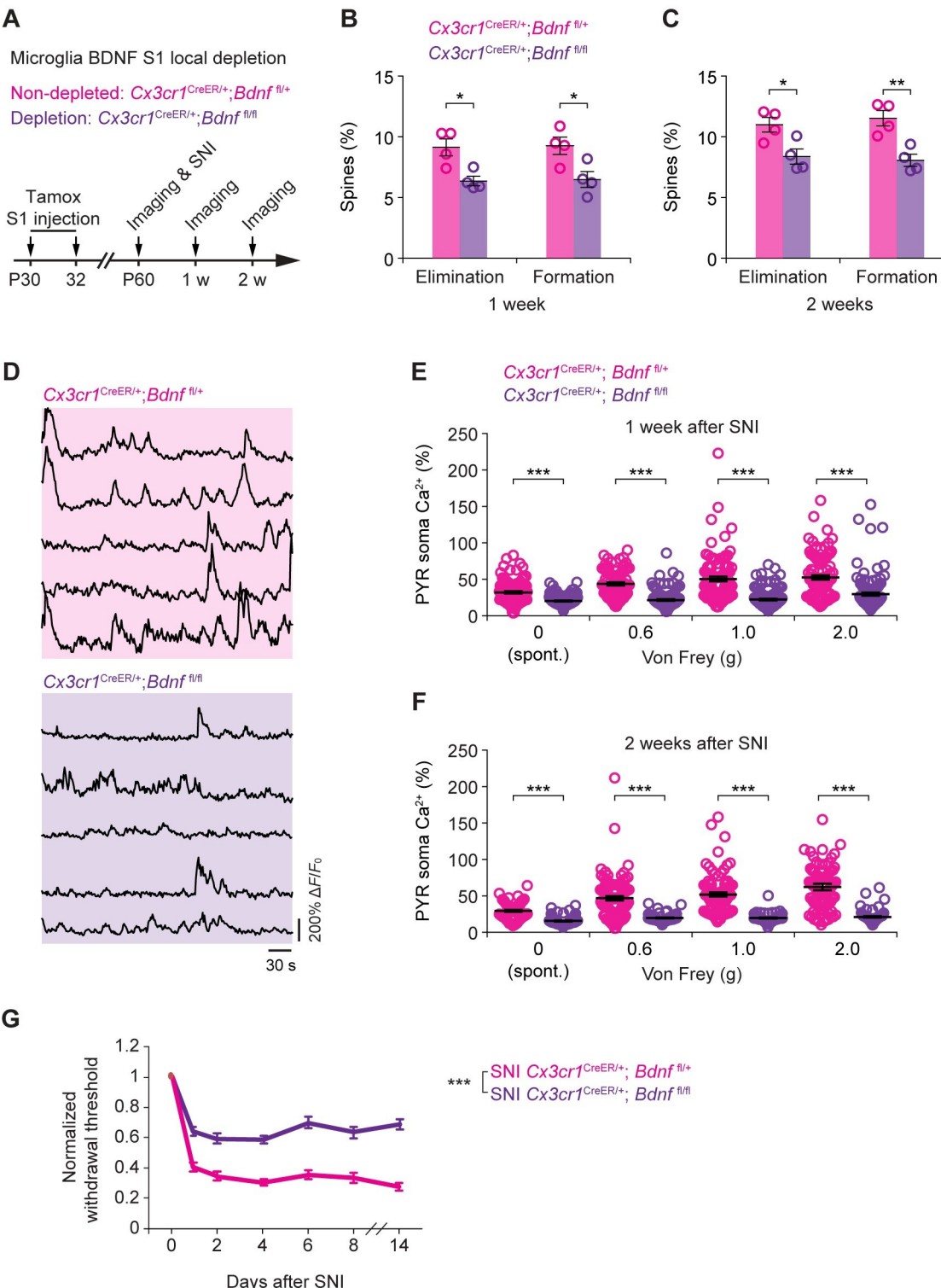

**Fig 8. S1-targeted microglial BDNF depletion reduces somatosensory cortical plasticity and allodynia.** (**A**) Experimental timeline. (**B, C**) Percentages of dendritic spine elimination and formation 1 week (**B**) or 2 weeks (**C**) after SNI in mice with or without S1 microglial BDNF ($Cx3cr1^{CreER/+}$;$Bdnf^{fl/+}$: 590 spines, $n = 4$ mice; $Cx3cr1^{CreER/+}$;$Bdnf^{fl/fl}$: 594 spines, $n = 4$ mice). (**D**) Representative fluorescence traces of L5 PYR somas expressing GCaMP6s in SNI mice with ($Cx3cr1^{CreER/+}$;$Bdnf^{fl/+}$) or without ($Cx3cr1^{CreER/+}$;$Bdnf^{fl/fl}$) S1 microglial BDNF. (**E**) Averaged Ca$^{2+}$ activity 1 week after SNI (0 g: $t_{218} = 7.323$, $P < 0.001$; 0.6 g: $t_{195} = 9.555$, $P < 0.001$; 1.0 g: $t_{230} = 9.464$, $P < 0.001$; 2.0 g: $t_{220} = 6.399$, $P < 0.001$). (**F**) Averaged Ca$^{2+}$ activity 2 weeks after SNI (0 g: $t_{154} =$

7.903, $P < 0.001$; 0.6 g: $t_{173} = 8.181$, $P < 0.001$; 1.0 g: $t_{172} = 10.21$, $P < 0.001$; 2.0 g: $t_{164} = 10.66$, $P < 0.001$). **(G)** Mechanical paw withdrawal threshold in mice with or without microglial BDNF in the S1 ($n = 10$ mice for each group; $F_{1,18} = 218.6$, $P < 0.001$). *$P < 0.05$, **$P < 0.01$, ***$P < 0.001$; by unpaired $t$ test (**B, C, E, F**), two-way ANOVA followed by Bonferroni post hoc test (**G**). The data underlying this figure can be found in S1 Data. BDNF, brain-derived neurotrophic factor; L5, layer 5; PYR, pyramidal; SNI, spared sciatic nerve injury; S1, primary somatosensory cortex.

increase of microglial *Bdnf*, *Tnf*-α, *Il*-6, and *Il*-1β occurred only in males, indicating a sex difference in brain microglial responses to peripheral nerve injury, which is consistent with recent transcriptomic findings that microglia in the healthy adult brain are sexually differentiated [40,41].

Following nerve injury or inflammatory insults in peripheral tissues, BDNF is up-regulated in the ipsilateral dorsal root ganglia, in the dorsal horn of the spinal cord, as well as in pain-processing brain regions such as the S1 and the ACC [42–46]. Although the major source of BDNF appears to be neurons, BDNF can also be detected in oligodendrocytes, astrocytes, and microglia [47]. Previous studies have shown that *Bdnf* transcript is at very low levels in microglia in both brain and spinal cord [26,27]. Accordingly, our RNAscope data show that *Bdnf* transcripts can only be visualized in a fraction of microglia in the S1, and most of these cells possess only a few *Bdnf* mRNA puncta under physiological conditions. Despite this low basal level of microglial *Bdnf* mRNA, we found that the count of *Bdnf* transcripts within microglia increased substantially after peripheral nerve injury. Moreover, the level of microglial *Bdnf* remains persistently elevated in males, consistent with the previous study that genetic deletion of microglial BDNF alleviates mechanical allodynia in male mice only [20].

BDNF acts as an activity-dependent neuromodulator and has potent effects on synaptic plasticity and neuronal network excitability [48–50]. Previous studies have shown that microglial BDNF is important for learning and learning-dependent synapse formation in the motor cortex [32]. In this study, we showed that BDNF derived from cortical microglia facilitates the maladaptive plasticity of cortical neurons after peripheral nerve injury. This was demonstrated by in vivo gene deletion experiments using *Cx3cr1*^CreER/+;*Bdnf*^fl/fl mice. Through conditional gene inactivation, we show that nerve injury–induced dendritic spine remodeling and pyramidal neuron hyperactivity in the contralateral S1 can be prevented either by removing BDNF from the entire CNS microglia population or by depleting BDNF selectively from microglia located within the S1 contralateral to the nerve injury side. Importantly, mice lacking microglial BDNF either across the CNS or within the S1 exhibit decreased mechanical allodynia after peripheral nerve injury. Together, these results underscore the importance of microglia-derived BDNF in the alterations of somatosensory cortical circuits as they relate to chronic neuropathic pain.

The mechanisms underlying microglial BDNF–dependent neuronal plasticity remain to be determined. In the spinal cord, nerve injury increases ATP release from dorsal horn neurons, which activates P2X4 receptors of spinal microglia, resulting in the phosphorylation and activation of p38–MAPK and subsequently the expression and release of BDNF [51–53]. Secreted BDNF binds neuronal TrkB receptors, leading to the down-regulation of potassium chloride cotransporter, which decreases the efficacy of GABA_A-mediated inhibition, resulting in the increased excitability of spinal neurons [21]. In the cortex, microglial processes are highly motile [54], located in close proximity to synaptic terminals, and have been implicated in synapse formation and pruning [55,56]. Under conditions of neuropathic pain, there is a substantial elevation of synaptic activity in apical dendrites of S1 pyramidal neurons [13]. Because ATP, a robust chemoattractant of microglial processes, can be released from nerve terminals [57], enhanced sensory input into the S1 may recruit local microglia processes via an ATP-dependent mechanism to synthesize and release BDNF. Microglial BDNF could facilitate

structural and functional remodeling of cortical neurons via similar mechanisms as demonstrated in the spinal cord. In addition, we found that microglia in the S1 up-regulate the expression of proinflammatory cytokines, including TNF-α, after peripheral nerve injury, and depletion of microglial BDNF prevents the increase of TNF-α. Given the important role of TNF-α in synaptic plasticity [58,59], it is also possible that cortical microglial BDNF promotes the neuronal plasticity in the S1 through the regulation of glial TNF-α.

Increasing evidence suggests that cortical circuits in the S1 actively contribute to the development of mechanical allodynia [11–14]. Because peripheral nerve injury results in maladaptive changes along the entire pain transmission pathway, it has been difficult to identify whether the changes observed in the S1 are simply a consequence of maladaptive changes in peripheral and spinal neurons or have an active role in pain chronicity. Through region-targeted and cell type–specific gene deletion strategy, we showed that the removal of BDNF solely from S1 microglia prevented structural and functional changes of S1 neurons after peripheral nerve injury. Importantly, these mice with reduced S1 plasticity after injury showed less mechanical allodynia. Because local injection of tamoxifen into the S1 does not affect microglia in other regions of CNS, including the spinal cord, these results provide direct evidence that local circuit changes within the S1 directly contribute to the development of mechanical allodynia after peripheral nerve injury. These results in mice resonate with the clinical findings that the strategies to reduce S1 hyperexcitation and reorganization show benefits against chronic pain [60,61], supporting a modulatory role of S1 in peripheral neuropathic pain.

In summary, our study identifies cortical microglia BDNF as a key regulator of cortical plasticity induced by peripheral nerve injury. The results provide direct evidence for the involvement of somatosensory cortical circuits in the pathogenesis of pain hypersensitivity. Although the precise signaling pathways underlying microglial BDNF function remain to be investigated in the SNI and other pain models, this finding might be of great interest for future studies aimed at developing innovative therapeutic strategies for chronic neuropathic pain by providing a valid alternative to harness neuron-glial communication.

## Materials and methods

### Ethics statement

All experimental protocols were conducted according to the National Institutes of Health (NIH) guidelines for animal research and approved by the Institutional Animal Care and Use Committee (IACUC) at New York University Medical Center (IACUC protocol # 16116–02) and Columbia University Medical Center (IACUC protocol # AAAW7462).

### Animals

*Thy1*-YFP mice (H line) were purchased from the Jackson Laboratory (Stock No: 003782). *Thy1*-GCaMP6s mice and *Cx3cr1*$^{CreER}$;*Bdnf*$^{flox}$ mice were generated in the laboratory of Dr. Wenbiao Gan [32,62]. For all experiments involving removal of microglial BDNF, *Thy1*-YFP and *Thy1*-GCaMP6s mice were crossed to *Cx3cr1*$^{CreER/+}$;*Bdnf*$^{fl/+}$ (nondepleted control) or *Cx3cr1*$^{CreER/+}$;*Bdnf*$^{fl/fl}$ (microglia BDNF–depleted mice). Mice were group housed in New York University Skirball animal facility and Columbia University Eye Institute animal facility.

### Spared nerve injury

SNI of the sciatic nerve or sham operation was performed under sterile conditions [30]. Adult mice (8 to 16 weeks) were deeply anesthetized, and the sciatic nerve and peripheral branches (common peroneal, tibial, and sural nerves) were exposed by a small incision in the left thigh.

The 8.0 nylon thread was slipped under the common peroneal and tibial nerves to make a ligation and cut. The nerve dissection was minimal, and any contact with the sural nerve was avoided, leaving the sural nerve intact.

## Electronic von Frey measurements

Mechanical pain threshold was measured using an Electronic von Frey Anesthesiometer, which consisted of a handheld force transducer fitted with a polypropylene tip (IITC, Life Science Instruments, USA). Mice were placed in acrylic cages ($12 \times 20 \times 17$ cm) with a mesh floor. During testing, the tip of the von Frey meter was applied perpendicularly to the mouse footpad with a gradual increase in pressure, until a flexion reflex was elicited. The pressure was automatically recorded when the paw withdrawal occurred. For each animal, the recording was repeated 3 times with more than 5 min in between, and the mean value was defined as the animal's mechanical withdrawal threshold. Von Frey tests were performed by the same researcher who was blind to the experimental conditions.

## Real-time quantitative PCR

One week after sham or SNI surgery, mice were perfused with PBS and S1 cortical tissues were collected. Microglia were isolated using fluorescence-activated cell sorting, and RNA was extracted from sorted microglia with an RNeasy Plus Mini Kit (Qiagen, USA). Extracted RNA (1 μg) was reverse transcribed into first-strand cDNA using Superscript II with Oligo(dT) (Life Technologies, USA). Quantitative reverse transcriptase PCR (qRT-PCR) was performed on an Opticon 2 thermal cycler (Biorad, USA) with FastStart Universal SYBR Green Master for the indicated genes (Roche, USA). The primers were designed and synthesized as follows: *Bdnf*, 5′-ACC CAT GGGATTACACTTGG-3′, 5′-AGCTGAGCGTGTGTGACAGT-3′; *Eef*2, 5′-TG TCAGTCATCGCCCATGTG-3′, 5′-CATCCTTGCGAGTGTCAGTGA-3′. *Il*-1β, 5′-TGGAGA GTGTGGATCC CAACAAT-3′, 5′-TGTCCTGACCACTGTTGTTTCCCA-3′; *Il*-6, 5′-CTGC AAGA GACTTCCATCCAGTT-3′, 5′-AAGTAGGGAAGGCCGTGGTT-3′; *Tnf*-α, 5′-TCGT AGCAAACCACCAAGTG-3′, 5′-CCTTGAAGAGAACCTGGGAGT-3′. The expression of *Bdnf* was normalized against *Eef*2. Data were derived from cells of 3 independent experiments from 8 mice for each group.

## RNAscope fluorescence in situ hybridization and data analysis

Fresh-frozen brains from *Cx3cr1*<sup>CreER-EYFP</sup> mice were sectioned into 12-μm thick slices. To visualize *Bdnf* or *Tnf*-α mRNA in sections of the S1, fluorescence in situ hybridization was performed according to manufacturers' instructions using the RNAscope Multiplex Fluorescent Reagent Kit v2 (#323100, Advanced Cell Diagnosis, USA). Prior to probe incubation, slices were pretreated with hydrogen peroxide (10 min, room temperature), Target Retrieval Reagent (5 min, 99°C), and RNAscope protease III (30 min, 40°C). Slices were incubated with *Bdnf* mRNA probes (#457761, Advanced Cell Diagnostics) targeting regions between bases 662 and 1,403 within the open reading frame or *Tnf*-α mRNA probes (#311081, Advanced Cell Diagnostics), and then hybridized to Opal 690 (PerkinElmer FP1497001KT) for visualization. Positive control probes (#312481, Advanced Cell Diagnostics) and negative control probes (#310043, Advanced Cell Diagnostics) were used in the parallel experiments to confirm specificity of hybridization. To visualize microglia together with mRNA puncta, brain slices were incubated at 4°C overnight with a primary antibody against GFP (1:250, Abcam, USA) and then for 2 h with Alexa Fluor 488-conjugated goat anti-rabbit IgG (1:500, Life Technologies, USA). After rinsing with PBS, the slices were incubated with DAPI solution (1:50,000) and mounted with ProLong Gold Antifade Mountant (Invitrogen, USA).

The images were acquired with a 40× objective at a zoom of 3.0 using a Zeiss LSM 800 confocal microscope. For each microscopic field of view (66 μm × 66 μm), a stack of 18 image planes at 0.5 μm increments along the z-axis were collected using the ZEN software, yielding a full three-dimensional data set. Microglia were identified from three-dimensional image stacks [32]. A total of 180 to 250 microglia from 4 animals per condition were examined for the presence of *Bdnf* mRNA, and the number of puncta per cell was quantified using the NIH ImageJ software. The negative controls were used to define the background during data analysis.

### In vivo imaging of dendritic spines and data analysis

The surgical procedure for performing transcranial two-photon imaging has been described previously [31]. In brief, mice were deeply anesthetized with 100 mg/kg ketamine and 15 mg/kg xylazine. After a midline incision in the scalp, a custom-made mounting plate with a central opening was glued to the skull surface with its opening right on top of the hindlimb region of S1 (0.5 mm posterior and 1.6 mm lateral from the bregma). A thinned-skull window (200 μm in diameter, 20 μm in thickness) was then created using a high-speed drill and a microsurgical blade. After that, the animal was placed under a two-photon microscope, and the laser was tuned to the optimal excitation wavelength of YFP (920 nm). A 60× water immersion objective (1.1 N.A.) was used to collect image stacks within a depth of 100 μm from the pial surface with a zoom of 1.0 to 3.0 and a resolution of 512 × 512 pixels. After imaging, the mouting plate was gently removed from the skull, and the scalp was sutured with 6–0 nylon. Mice were returned to their home cages until the next viewing.

Data analysis was performed with the NIH ImageJ software as described previously [63]. The same dendritic segments were identified from the three-dimensional data sets collected at different time points. The number and location of dendritic protrusions were identified in each view and compared between views. Filopodia were classfied as long and thin structures without enlarged heads. The rest of dendrtic protrusions were categorized as spines. Spines in the second view were considered different if they were >0.7 μm away from their expected position in the first view on the basis of their spatial relationship to nearby spines and local landmarks. The formation or elimination rates of spines were quantified as the number of spines formed or eliminated divided by the number of spines existing in the first view.

### In vivo Ca$^{2+}$ imaging and data analysis

Ca$^{2+}$ imaging was performed in awake, head-restrained mice [34]. To enable the head-fixation, a custom-made head holder was permanently attached to the animal's skull using both glue and dental acrylic cement while the mouse was under deep anesthesia. After the cement was dry, a cranial window was created over the hindlimb region of S1. The procedure for preparing a thinned-skull window for two-photon imaging has been described in detail in previous publications [31]. Upon completion, the window was covered with silicone elastomer and mice were returned to their home cages to recover for at least 24 h. Before imaging, mice with the head holder were habituated to a mounting device 3 times (10 min each) to minimize the potential stress related to the head-fixation.

During imaging, the head holder was screwed to the mounting device and the silicon elastomer was peeled off to expose the cranial window. The Ca$^{2+}$ imaging experiments were performed using an Olympus two-photon system equipped with a DeepSee Ti:sapphire laser (Spectra Physics, USA). Images were collected at a frame rate of 2 Hz and a resolution of 256 × 512 pixels using a 25× water immersion objective (1.05 N.A.). Image acquisition was performed using Olympus FV1000 software and analyzed post hoc using the NIH ImageJ software. The fluorescence time course was measured by averaging all pixels within the regions of

interest covering a soma after the correction of background. $\Delta F/F_0$ was calculated by $(F - F_0) / F_0$, where $F_0$ is the baseline fluorescence signal averaged over a 20-s period corresponding to the lowest fluorescence signal during the 2.5-min recording period. The mean AUC (area under the curve) over 2.5 min was presented as the somatic $Ca^{2+}$ activity under various conditions. To determine stimulation-evoked neuronal responses, a von Frey filament was applied to the plantar surface of mouse paw for 10 s, and $Ca^{2+}$ recording during this period was divided into 10 bins equally. A z-score for each bin was calculated relative to 10 bins of prestimulation period. Neurons were classified as showing increased responses if any stimulation bin exceeded 2. Neurons were classified as showing decreased responses if any stimulation bin exceeded −2.

## Tamoxifen-induced recombination

All $Cx3cr1^{CreER/+}$ mice received tamoxifen on P30 and P32. For systemic administration, tamoxifen (Sigma, USA) was dissolved in corn oil (Sigma, USA) and given to mice by oral gavage. Animals received 2 doses of 10 mg tamoxifen with 48 h in between. For intracranial administration, 4-OHT was solubilized in Cremophor EL (Sigma, USA) and delivered to the mouse cortex by stereotaxic microinjection. A total of 200 μM 4-OHT was diluted 10 times in artificial cerebrospinal fluid and slowly injected (Picospritzer III; 15 p.s.i., 10 ms, 0.5 Hz) over 30 to 45 min into L5 (depth of 500 to 800 μm) of the S1 using a glass microelectrode around the coordinates of 0.5 mm posterior and 1.8 mm lateral from the bregma.

## Statistics

Prism software (GraphPad 7.05) was used to conduct the statistical analysis. Data were presented as mean ± SEM. Tests for differences between populations were performed using a two-tailed Student $t$ test or a one- or two-way ANOVA followed by Bonferroni or Tukey tests as specified in the text. Significant levels were set at $P < 0.05$.

## Supporting information

**S1 Fig. Confocal z-stack images showing *Bdnf* mRNA within the microglia.** Orthogonal sections from z-stack confocal images of microglia (green), *Bdnf* mRNA (red), and DAPI (blue) in sham (left) and SNI (right) groups. *Bdnf*, brain-derived neurotrophic factor; SNI, spared sciatic nerve injury.
(TIF)

**S2 Fig. Peripheral nerve injury has no effect on dendritic spine density in the S1. (A)** The net change of total spine number in the S1 at various time points after sham or SNI surgery. (**B**) Density of dendritic spines on the apical tuft dendrites of L5 pyramidal neurons from SNI and sham mice (sham: 1074 spines, $n = 7$ mice; SNI: 1,106 spines, $n = 7$ mice). Data are presented as means ± SEM unpaired $t$ test. The data underlying this figure can be found in S1 Data. L5, layer 5; SNI, spared sciatic nerve injury; S1, primary somatosensory cortex.
(TIF)

**S3 Fig. Absence of *Bdnf* mRNA in microglia in tamoxifen-treated $Cx3cr1^{CreER/+};Bdnf^{fl/fl}$ mice. (A)** RNAscope fluorescence in situ hybridization in the S1 of $Cx3cr1^{CreER/+};Bdnf^{fl/fl}$ mice. Red, *Bdnf* mRNA probe hybridization. Green, $Cx3cr1$-EYFP$^+$ microglia. Blue, DAPI. Scale bar, 20 μm. (**B**) Normalized levels of *Bdnf* mRNA in microglia ($n = 3$ mice per group). $^{**}P < 0.01$, $^{***}P < 0.001$; by one-way ANOVA followed by Bonferroni multiple comparisons test. The data underlying this figure can be found in S1 Data. *Bdnf*, brain-derived neurotrophic

factor; S1, primary somatosensory cortex; S1HL, hindlimb region of S1.
(TIF)

**S4 Fig. SNI-induced microglia *Tnf-α* expression was abolished by depletion of microglial BDNF.** (**A**) RNAscope fluorescence in situ hybridization in the S1 of sham and SNI mice with or without microglial BDNF. (**upper**) Sham *Cx3cr1*<sup>CreER/+</sup>;*Bdnf*<sup>fl/+</sup>, (**middle**) SNI *Cx3cr1*<sup>CreER/+</sup>;*Bdnf*<sup>fl/+</sup>, (**bottom**) SNI *Cx3cr1*<sup>CreER/+</sup>;*Bdnf*<sup>fl/fl</sup>. Red color represents *Tnf-α* mRNA probe hybridization. Green color indicates *Cx3cr1*-EYFP<sup>+</sup> microglia. Blue, DAPI. Scale bar, 10 μm. (**B**) Normalized levels of *Tnf-α* mRNA in microglia ($n = 3$ mice per group). $^*P < 0.05$. $^{**}P < 0.01$; by one-way ANOVA followed by Bonferroni multiple comparisons test. The data underlying this figure can be found in S1 Data. BDNF, brain-derived neurotrophic factor; SNI, spared sciatic nerve injury; S1, primary somatosensory cortex.
(TIF)

**S5 Fig. Enhanced sensory-evoked neuronal Ca<sup>2+</sup> activity in the S1 of mice with neuropathic pain.** (**A, B**) Averaged somatic Ca<sup>2+</sup> activity over 10 s in S1 PYR neurons before and during mechanical stimulation by 1 g (**A**) or 2 g (**B**) von Frey hair at 1 week after surgery (sham: 5 mice; SNI: 8 mice). (**C–E**) Averaged somatic Ca<sup>2+</sup> activity over 10 s in S1 PYR neurons before and during 0.6 g (**C**), 1 g (**D**), or 2 g (**E**) paw stimulation at 2 weeks after surgery (sham: 5 mice; SNI: 5 mice). Throughout, individual circle represents data from a single cell. $^{***}P < 0.001$, ns, not significant; by paired *t* test. The data underlying this figure can be found in S1 Data. PYR, pyramidal; SNI, spared sciatic nerve injury; S1, primary somatosensory cortex.
(TIF)

**S1 Data. Data underlying Figs 1A, 1C, 1E, 1F, 2D, 2E, 3C–3F, 4C–4E, 5D, 5F, 5H, 5I, 6C–6E, 7D, 8B, 8C, 8E–8G, S2A, S2B, S3B, S4B, and S5.**
(XLSX)

## Acknowledgments

We thank Gan lab and Yang lab members for helpful discussions.

## Author Contributions

**Conceptualization:** Lianyan Huang, Wen-Biao Gan, Guang Yang.

**Data curation:** Lianyan Huang, Wen-Biao Gan, Guang Yang.

**Investigation:** Jianhua Jin, Kai Chen, Sikun You, Hongyang Zhang, Guang Yang.

**Methodology:** Lianyan Huang, Jianhua Jin, Kai Chen, Sikun You, Hongyang Zhang, Alexandra Sideris, Monica Norcini, Esperanza Recio-Pinto, Jing Wang, Wen-Biao Gan, Guang Yang.

**Supervision:** Lianyan Huang, Guang Yang.

**Validation:** Kai Chen, Sikun You.

**Writing – original draft:** Lianyan Huang, Guang Yang.

**Writing – review & editing:** Lianyan Huang, Jianhua Jin, Kai Chen, Sikun You, Hongyang Zhang, Alexandra Sideris, Monica Norcini, Esperanza Recio-Pinto, Jing Wang, Wen-Biao Gan, Guang Yang.

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
