## [Editor Report · Decision Letter 0]

24 Aug 2020

Dear Dr Huang, 

Thank you for submitting your manuscript entitled "Cerebral microglial BDNF promotes peripheral nerve injury-induced cortical plasticity and pain hypersensitivity" for consideration as a Research Article by PLOS Biology.

Your manuscript has now been evaluated by the PLOS Biology editorial staff as well as by an academic editor with relevant expertise and I am writing to let you know that we would like to send your submission out for external peer review.

Please re-submit your manuscript within two working days, i.e. by Aug 26 2020 11:59PM.

Kind regards,

Lucas Smith, Ph.D.,

Associate Editor

PLOS Biology

---

## [Decision Letter · Decision Letter 1]

15 Oct 2020

Dear Dr Huang,

Thank you very much for submitting your manuscript "Cerebral microglial BDNF promotes peripheral nerve injury-induced cortical plasticity and pain hypersensitivity" for consideration as a Research Article at PLOS Biology. Your manuscript has been evaluated by the PLOS Biology editors, an Academic Editor with relevant expertise, and by several independent reviewers.

The reviews of your manuscript are appended below. As you will see from their detailed responses, the reviewers agree that the study presents potentially interesting findings, but argue that the findings need to be bolstered with additional data and analyses to be suitable for publication in PLOS Biology. In particular, all three reviewers raise concerns with the sole use of male mice, given the known sexual dimorphism of microglial BDNF in neuropathic pain. Having discussed the reviewers’ comments with an Academic Editor, we appreciate that the request to perform additional experiments in female mice is a large undertaking, however, we feel that adding such analyses is essential. Reviewers 1 and 2 additionally raised concerns with the RNAscope analysis in Figure 2, and Reviewer 2 asked for further analyses of other signaling molecules.

We will not be able to accept the current version of the manuscript, but we would welcome re-submission of a much-revised version that addresses the reviewers' comments. We cannot make any decision about publication until we have seen the revised manuscript and your response to the reviewers' comments. Your revised manuscript is also likely to be sent for further evaluation by the reviewers.

We expect to receive your revised manuscript within 3 months. However if you feel that some experiments may take longer than that, especially given the COVID-19 pandemic, and we would be happy to extend the time for revision. 

**IMPORTANT - SUBMITTING YOUR REVISION**

*Re-submission Checklist*

*Published Peer Review*

*PLOS Data Policy*

*Blot and Gel Data Policy*

Sincerely,

Lucas Smith, Ph.D.,

Associate Editor,

lsmith@plos.org,

PLOS Biology

REVIEWS:

Reviewer's Responses to Questions

PLOS authors have the option to publish the peer review history of their article (what does this mean?). If published, this will include your full peer review and any attached files.

Reviewer #1: No

Reviewer #2: No

Reviewer #3: Yes: Marie-Eve Tremblay

Reviewer #1: This is a very interesting paper to investigate the role of microglial BDNF in the S1 sensory cortex for neuropathic pain after spared nerve injury (SNI). The authors used several cutting-edge techniques such as in vivo two-photon imaging of morphological changes (dendrites) and functional changes (calcium increase) and Cx3cr1CreER:Bdnfflox mice. The data showed that conditional knockout of BDNF from microglia can prevent nerve injury-induced synaptic remodeling and pyramidal neuron hyperactivity in the S1, as well as pain hypersensitivity in mice. Finally, the authors showed that S1 targeted removal of microglial BDNF via intracranial administration of tamoxifen could largely reproduce the beneficial effects of systemic BDNF depletion on cortical plasticity and mechanical allodynia. Notably, the authors examined the morphology, cell density and process motility of microglia in the S1 after nerve injury and observed no differences in the morphology and density of microglia in the S1 between SNI and sham mice (Fig 1B, C). This result is important to suggest that microglia can regulate pain even in the absence of obvious morphological changes. Overall, this is well conducted study, and the findings are significant. The following issues should be addressed before the paper is accepted for publication. 

1) The RNAscope data for BDNF in figure 2 is underwhelming. There seems to be a change in BDNF mRNA between sham and nerve injury, but it seems like most of the signal is outside of microglia and it is difficult to see if the signal is above or below the microglial cells. Please describe the way the images are acquired. It will be helpful to show a confocal Z-stack to show the RNA puncta within the microglia. 

2) It is difficult to see Bdnf mRNA expression by 60% of microglia. In Fig. 2A with SNI, it looks only one microglial cell showing clear puncta in a field of 5 microglia. 

3) If you only have 2 puncta per microglial cell, how is the background defined for RNAscope? I wonder if you did RNAscope using Cx3cr1CreER/+:Bdnffl/fl mice. 

4) Fig. 2E. It is unclear how many animals were used for the quantification. How many sections were evaluated per animal? I wonder if the sample size is based on the number of sections or number of microglia instead of number of animals? 

5) Male mice were used in this study. Spinal microglial BDNF signaling was shown to be male dominant in neuropathic pain. I wonder if there is sex difference in cortical microglia. 

Other comments:

Discussion: "Following nerve injury or inflammation in peripheral tissues, BDNF is upregulated in the ipsilateral dorsal root ganglia, in the dorsal horn of the spinal cord, as well as in supraspinal regions and various cortical areas such as the S1 and the ACC (ref 37-39)". The authors may check earlier studies showing BDNF upregulation in DRG neurons in inflammatory pain and neuropathic pain (PMID: 10430952; PMID: 11425916).

Discussion: "targeting microglia-neuron signaling pathways in the somatosensory circuits could be an effective strategy to prevent and treat chronic neuropathic pain". It will be difficult to target this signaling pathway in the S1 cortex without causing side effects. This brain region is also difficult to access using non-invasive method. BDNF also has beneficial effects. Additional discussion will help the paper. 

Reviewer #2: The present investigation by Huang et al was undertaken to identify a role of microglia in abnormal cortical plasticity associated with the spared nerve injury (SNI)-induced neuropathic pain. In this study, authors noted no change in the cortical microglia morphology and density, and a mild change in the motility of microglia processes post-SNI. Also, it was reported that the expression of Bdnf mRNA was upregulated in microglia of the contralateral primary somatosensory cortex (S1), assessed using qPCR and RNAscope fluorescence in situ hybridization. In a series of experiments involving the use of transgenic mice and two-photon imaging authors observed that the SNI mice show an increase in synapse remodeling (increase in elimination and formation of dendritic spines) and somatic Ca2+ responses in L5 neurons. However, upon genetic depletion of BDNF, synaptic remodeling, neuronal hyperactivation and mechanical allodynia was prevented. Based on the findings presented here, authors propose that microglia-neuron signaling involving BDNF is important for abnormal cortical plasticity pertaining to chronic pain caused by traumatic peripheral nerve injury. 

Major Comments to the Authors:

1. Microglial BDNF involvement in SNI pain behavior has been reported to be restricted to male specific pain pathways (Sorge 2015, Zhou 2019, Saika 2020). Therefore the authors must test the necessity of microglial BDNF in S1 cortex in female mice with SNI. If there is no sex difference, this would be a highly intriguing finding. The Authors should also consider local tamoxifen-induced knockout of BDNF in the spinal cord to ascertain if spinal microglial-derived BDNF is dispensable for pain behaviors in male and female mice. 

2. Molecular assessment was not performed on tamoxifen treated CX3CR1-BDNF(fl/fl) mice to determine whether Bdnf mRNA was indeed depleted. As in Figure 1F, the authors need to perform qPCR for Bdnf expression in isolated microglia from the knockout mice. Additionally, the Authors need to measure IL-6, IL-1beta and TNF-alpha as well in the knockout cells. Liu 2017 has shown that brain TNF-alpha from microglia regulates synaptic plasticity and spine formation after SNI, and Rossetti 2019 has shown that IL-1beta in some parts of the brain increases in expression in females after BDNF knockout. Therefore it is necessary for the Authors to assess if microglia from tamoxifen treated CX3CR1-BDNF(fl/fl) mice express these signaling transcripts differentially as any of these molecules may provide a mechanistic explanation for the observed dendritic spine remodeling.

3. The RNAscope assessment of Bdnf mRNA puncta is not convincing. The Bdnf mRNA puncta are distributed across the images but very few are localized over microglia somata. The'yellow' puncta could represent neuronal, or other, sources overlapping the soma of the microglia. The Authors need to show the side profiles for these images to show the readers whether the signals are present inside the soma of microglia. Additionally, the Authors need to perform RNAscope in brain sections from the microglia Bdnf knockout. It is surprising this was not done as it would provide irrefutable evidence supporting the validity of using RNAscope to detect Bdnf mRNA in microglia.

4. Regarding somatic Ca2+ recordings. The figure panels for the analyses unclear. For example, in figure 5C, it is not clear how percent change is calculated given that the 'baseline' is changing. Which part of these traces are used as baseline? Is the measured change the distance of peaks vs baseline? Why not calculate area under the curve? In the figure 5D, does 0% mean no change from baseline (ie, 100% equal to baseline)? For evoked Ca2+ measurements, it is mentioned in the method that the 10s prior to evoked stimulation is used as baseline. However, in all the figures showing raw traces during evoked stimulation, the baselines are cut off. The Authors should show at minimum a full 20s of traces prior to the evoked stimulation so that the readers can see what the baseline looked like. Area under the curve would seem to be a better measure of Ca2+ signal, doesn't require establishing a baseline and is easier to interpret.

5. Regarding Figure 8D,E. The Authors need to show spontaneous Ca2+ traces akin to Figure 6. Did local S1 depletion of Bdnf affect spontaneous Ca2+ transients in S1 neurons? 

Other comments to the Authors:

1. Typo (text says 'BNDF' - it should be 'BDNF') under intro section - paragraph 3, results fig 7 description 

2. Regarding Figure 1F. It is well known that activity of microglia affects expression of actin and actin regulatory molecules. The use of actin mRNA as a single reference for microglia gene expression is questionable. It is commonly accepted to use Hprt1, 18S, Eef2 or Abt1 as reference genes for microglia/macrophage cells. The authors should consider repeating the experiments in figure 1F with a more appropriate reference gene(s). Additionally, the figure includes the double normalized Sham groups (ddCT method). This is unnecessary and only the SNI can be displayed with a dashed line across the value of 1 to depict the doubly normalized sham values. Ideally the authors should display only the dCT values (singly normalized) for both the sham and SNI groups in order to show the readers the degree of variation present in the Sham groups and also the degree of detectability for each of the transcripts.

3. Regarding Figure 7B. Why is there no background signal in spinal cord DsRed panel? It seems the exposure was set far too low. Also, Figure 7c, the spinal cord section seems to be a thoracic segment, or at least does not appear to be L4/5 level which is expected to be shown for a hindpaw SNI model.

4. Regarding Figure 8. If i.t. injection of tamoxifen can deplete spinal microglial Bdnf, then the Authors should test pain behavior to ascertain if local spinal microglial Bdnf is required for pain behavior. This would add considerably to establishing necessity versus sufficiency of S1 microglial Bdnf involvement in SNI pain signaling.

Reviewer #3: This study by Huang et al. reveals that microglia play an important role in the structural and functional changes taking place in the primary somatosensory cortex (S1) after peripheral nerve injury in mouse. Following peripheral nerve injury, microglia in the S1 maintain a surveillant morphology and normal density but they up-regulate their mRNA expression level of brain-derived neurotrophic factor (BDNF). Using transcranial in vivo two-photon imaging and Cx3cr1CreER:Bdnfflox mice, the authors show that conditional knockout of BDNF from microglia prevents nerve injury-induced synaptic remodelling and pyramidal neuronal hyperactivity in the S1, as well as pain hypersensitivity in mice. In addition, S1 targeted removal of microglial BDNF largely recapitulated the beneficial effects of systemic BDNF depletion on cortical plasticity and allodynia. Together, these findings reveal an unexpected role of cerebral microglial BDNF in somatosensory cortical plasticity and pain hypersensitivity.

The study uses cutting-edge mouse models as well as technical approaches. It is elegantly designed and conducted.

I only have a few important requests for revision to strengthen the findings:

1) The rationale for performing experiments in male mice only should be explained, especially considering the important sexual dimorphism in microglial BDNF function that was reported (e.g. in models of neuropathic pain).

2) Throughout the manuscript, the microglial nomenclature should be revised based on the seminal findings, notably from co-author Wenbiao Gan, that microglia are not quiescent at steady state. Ramified or quiescent should thus be replaced by surveillant.

3) The LSD posthoc test is not stringent enough and it would be strongly recommended to use Bonferroni or Tukey instead. 

4) The discussion is the weakest part of this manuscript. It would be important to elaborate further on the limitations of the study, including models used (e.g. Thy1-YFP mice display different responses to spinal injury versus wild-type mice; tamoxifen modulates the immune system including microglia)

5) It would be interesting to provide further discussion on how, by which mechanisms downstream of BDNF, microglia modulate the response to peripheral nerve injury.

---

## [Decision Letter · Decision Letter 2]

27 May 2021

Dear Dr Huang,

Thank you for submitting your revised Research Article entitled "Cerebral microglial BDNF promotes peripheral nerve injury-induced cortical plasticity and pain hypersensitivity" for publication in PLOS Biology. I have now obtained advice from the original reviewers and have discussed their comments with the Academic Editor. 

As you will see from their comments, the reviewers are satisfied with your revision and raise no additional concerns. Therefore, we will probably accept this manuscript for publication, provided you satisfactorily address the following data and other policy-related requests, outlined here:

1) FINANCIAL DISCLOSURES: Where possible, please provide the URL for each funder's website. 

2) ETHICS REQUEST: Please note the approval number(s) of the animal use and care protocols approved by the IACUC at NYU and Columbia. Please also make sure to include the specific national or international regulations/guidelines to which your animal care and use protocol adhered. More information about this request can be found below my signature. 

3) DATA REQUEST: Please provide as a supplementary file, or as a deposition in a publicly available repository, the data underling each figure in your manuscript. **Please also make sure to reference this file in each figure legend (including supplemental). For example, to each figure legend you might add a sentence saying "the data underlying this figure can be found in S1_Data." Please also ensure that this file has a legend. More information about this request can be found below my signature. 

4) Having discussed the title of your manuscript within the team, we wonder if it might be edited slightly to improve its clarity. If you agree, we might suggest something like "BDNF produced by cerebral microglia promotes cortical plasticity and pain hypersensivity after peripheral nerve injury". However, we will ultimately leave it to you if and how to change the title. 

We expect to receive your revised manuscript within two weeks. 

*Published Peer Review History*

*Early Version*

Sincerely,

Lucas Smith, Ph.D.,

Associate Editor,

lsmith@plos.org,

PLOS Biology

ETHICS STATEMENT:

-- Please the approved number of the animal care and use protocol/permit/project license approved by the IACUCS at NYU and Columbia.

-- Please include the specific national or international regulations/guidelines to which your animal care and use protocol adhered. Please note that institutional or accreditation organization guidelines (such as AAALAC) do not meet this requirement.

DATA POLICY:

Fig 1A,C,E-F; Fig 2D-E; Fig 3C-F; Fig 4 C-E; Fig 4D-I; Fig 6C-E; Fig 7D; Fig 8 B-C,E-G; Fig S2A-B; Fig S3B; Fig S4B; Fig S5.

**IMPORTANT: Please also ensure that figure legends in your manuscript include information on where the underlying data can be found, and ensure your supplemental data file/s has a legend.

**IMPORTANT: Please ensure that your Data Statement in the submission system accurately describes where your data can be found.

Reviewer remarks:

Reviewer #1: The authors have conducted additional experiments and included new data to show 1) detailed analysis of Bdnf expression in microglia and 2) sex-dependent changes in cortical microglia. I have no further comments. 

Reviewer #2: The authors have addressed a sufficient number of my concerns to make the paper acceptable. 

Reviewer #3: Thank you for your revision, which addresses all my concerns. I am happy to recommend publication of your manuscript.

---

## [Editor Report · Decision Letter 3]

22 Jun 2021

Dear Dr Huang,

On behalf of my colleagues and the Academic Editor, Mikael Simons, I am pleased to say that we can in principle offer to publish your Research Article "BDNF produced by cerebral microglia promotes cortical plasticity and pain hypersensitivity after peripheral nerve injury" in PLOS Biology, provided you address any remaining formatting and reporting issues. These will be detailed in an email that will follow this letter and that you will usually receive within 2-3 business days, during which time no action is required from you. Please note that we will not be able to formally accept your manuscript and schedule it for publication until you have made the required changes.

As you address these last requests, we also ask that you add to your methods section information on the the national or international regulations/guidelines to which their animal care and use protocol adhered. Please note that institutional or accreditation organization guidelines (such as AAALAC) do not meet this requirement. An example of guidelines that would fit this category would be the NIH Guide for the care and use of laboratory animals.

PRESS

Sincerely, 

Lucas Smith, Ph.D. 

Associate Editor 

PLOS Biology

lsmith@plos.org